# Advances in Glioblastoma Diagnosis: Integrating Genetics, Noninvasive Sampling, and Advanced Imaging

**DOI:** 10.3390/cancers17010124

**Published:** 2025-01-02

**Authors:** Ryan Gough, Randall W. Treffy, Max O. Krucoff, Rupen Desai

**Affiliations:** Department of Neurosurgery, Medical College of Wisconsin, Milwaukee, WI 53226, USA; rgough@mcw.edu (R.G.); rtreffy@mcw.edu (R.W.T.); maxkrucoff@mcw.edu (M.O.K.)

**Keywords:** glioblastoma, glioblastoma genetics, glioblastoma liquid biopsy, glioblastoma-targeted therapy, MR spectroscopy

## Abstract

Advances in molecular techniques have significantly improved our understanding of glioblastoma in recent decades. Here, we describe the genetic mutations important in the pathogenesis, diagnosis, prognosis, and treatment of glioblastoma. We then describe novel diagnostic tools that allow for faster and less invasive diagnosis as well as monitoring of treatment response and disease recurrence. This summary will serve as a reference for the current state-of-the-art and future directions for glioblastoma treatment.

## 1. Introduction

Glioblastoma is the most common primary brain tumor in adult patients, and despite standard-of-care therapy with maximal safe surgical resection, chemotherapy, and radiotherapy, prognosis remains grim, with a median survival of less than 2 years [1,2,3,4]. While glioblastoma can occur anywhere within the central nervous system (CNS), it is most commonly localized to the frontal lobe or temporal lobe [4], likely due to disproportionate tissue volumes in these regions [1,2,3]. Previous WHO classifications primarily comprised histological descriptions of hypercellularity, nuclear atypia with increased mitotic activity, and microvascular proliferation or pseudopalisading necrosis [5]. However, recent innovations in genomic techniques have revealed some of the key drivers of glioblastoma pathogenesis [6]. The 2021 iteration of the WHO glioblastoma classification incorporates some of these vital molecular analyses given significant clinical differences between molecular subgroups [6].

Given the constantly changing landscape of glioblastoma, in this review, we describe the current state of glioblastoma genomics and potential implications of ongoing studies. We additionally explore alternative, less invasive techniques being evaluated for glioblastoma diagnosis and treatment monitoring.

## 2. Molecular Profiling of Glioblastoma

As with other fields of oncology, there have been dramatic advances in our understanding of glioblastoma with advances in high-throughput sequencing technology. As mentioned before, the classic diagnostic criteria for glioblastoma was based on pathological examination of the sample taken from surgical resection or biopsy [5]. However, as our ability to probe the genetic drivers of glioblastoma has advanced, we have realized that there are key genetic mutations that are critical to the pathogenesis of glioblastoma and serve as key molecular markers of overall prognosis and response to therapeutic intervention [6]. Therefore, we will explore common and important genetic alterations, their clinical significance, and potential treatments associated with those genetic findings.

### 2.1. Isocitrate Dehydrogenase

Isocitrate dehydrogenase (IDH) is an enzyme that catalyzes the reversible oxidative decarboxylation of isocitrate to alpha-ketoglutarate within the Krebs cycle, a cellular metabolic pathway [7]. Clinically relevant mutations have been described in both IDH1 (cytosolic localized) and IDH2 (mitochondrial localized) isoforms and most frequently occur at arginine 132 in IDH1 or arginine 172 or 140 in IDH2 [7]. The relevance of IDH mutations in glioblastoma was first described in 2008, with IDH1 mutations found to be more common in younger patients and associated with an increase in survival [8].

The causal relationship between IDH mutation and glioma pathogenesis is unclear, although it is speculated that IDH mutations result in reduced alpha-ketoglutarate function, with downstream effects on both epigenetic cellular metabolism alterations [7]. IDH mutations are powerful driver mutations, as a single mutation is sufficient to drive glioma hypermethylation [9] and prevent histone demethylation, with implications for cellular terminal differentiation [10]. Interestingly, while IDH mutation is associated with improved outcomes in glioma, these epigenetic changes associated with mutant IDH lead to increased proliferation [11]. Further mechanistic studies of the IDH mutation are necessary to better understand exactly how this mutation leads to glioma formation.

Given IDH mutations are associated with improved outcomes [12,13], the 2021 WHO guidelines made significant alterations to the pathologic classification. Rather than relying exclusively on histologic characteristics, the current classification requires wild-type IDH status to achieve a glioblastoma diagnosis [6,13]. Importantly, IDH status has only recently been evaluated in glioblastoma clinical trials and even individual patient pathologic annotations; thus, historic comparisons should be evaluated with this caveat.

Determining IDH status is important and can be performed via immunohistochemistry (IHC) or via sequencing. Typically, the first step is to use IHC, which specifically targets the R132H mutation and does not detect other potential, less common mutations in IDH [14]. If IHC testing is negative, the next step is typically sequencing, which would be more accurate but is more costly and time-consuming [15].

Importantly, targeted therapy for IDH has recently been evaluated, demonstrating promise in patients with IDH mutations. The INDIGO trial demonstrated that vorasidenib, an IDH inhibitor, improved survival in those with residual or recurrent grade 2 IDH-mutant glioma [16]. While a survival benefit was not demonstrated in patients with contrast-enhancing tumors (generally indicative of higher-grade lesions) [17], there is promising data in this population with the IDH inhibitor olutasidenib [18]. As these studies continue to evolve, the outcome for those with higher-grade tumors treated with IDH inhibitors will become clearer. As of now, the presence or absence of an IDH mutation is critically important in those with glioma, and according to the WHO 2021 classification, it excludes the diagnosis of glioblastoma.

### 2.2. MGMT

O6-methylguanine-DNA methyltransferase (MGMT) is a critical clinical biomarker for those diagnosed with glioblastoma [19]. MGMT is an important component of the DNA repair pathway and mediates repair of O6-alkylguanine lesions, which can be caused by environmental carcinogens and chemotherapeutic agents, in particular alkylating agents [20,21]. Temozolomide, the current standard-of-care chemotherapy for glioblastoma, is one such DNA alkylating agent, and it was suspected that MGMT dysfunction may improve therapeutic efficacy. Indeed, it was found that inactivation of MGMT by methylation of its promoter leads to improved clinical response of gliomas to alkylating agents [22] and specifically to temozolomide [23].

Interestingly, while MGMT activity can help predict the sensitivity of glioblastoma to temozolomide chemotherapy, current assays are imperfect. Currently, MGMT status is most commonly assessed by evaluating the extent of the methylation of the promoter region; however, while there is a strong correlation between promoter methylation and MGMT activity, some variability is notable [19,24]. Furthermore, there is no standardized cut-off used in clinical practice when comparing MGMT methylation status, which may further complicate the interpretation of MGMT function [19,25,26,27].

Yet another complicating factor is the induced expression of MGMT caused by temozolomide treatment. Treatment of glioblastoma cell lines with temozolomide in vitro led to a significant increase in expression of MGMT, leading to resistance to temozolomide treatment [28]. Furthermore, it is likely that significant intra-tumoral MGMT heterogeneity exists [29]. Thus, both the induction of MGMT and the selective pressure by temozolomide may lead to temozolomide resistance [28,29].

Although MGMT methylation status is not used in the integrated diagnosis of glioblastoma, it remains a critical prognostic tool as there are significant variations in clinical response to standard-of-care treatments (temozolomide) based on the status of MGMT. Importantly, as patients with unmethylated MGMT have diminished response to standard-of-care temozolomide therapy, this has been an important factor for inclusion in clinical trials for glioblastoma.

### 2.3. TERT

Telomeres are the linear ends of chromosomes involved in cellular replication, allowing complete replication of the coding portions of the chromosome [30]. In this process, telomeres shorten with each cellular replication, triggering apoptosis after reaching a threshold [30]. Telomerase reverse transcriptase (TERT) is an important enzyme required for telomere lengthening [30] and is often subverted by malignant cells, particularly in otherwise quiescent organs, to promote telomere lengthening [31]. TERT promoter mutations occur in two well-described hotspots (C228T and C250T), which improve binding by the GA-binding protein transcription factor alpha subunit and increase TERT mRNA [32]. This increase in TERT mRNA leads to an increase in telomerase activity and can lead to cellular immortalization and tumor progression [33].

TERT promoter mutations are found in about 80% of glioblastoma patients and are typically diagnosed by DNA sequencing [34]. Interestingly, telomere length is not significantly increased by TERT promoter mutations [35], suggesting a TERT promoter mutation is likely a late mutation. However, other data suggest that TERT promoter mutations may occur early in glioblastoma pathogenesis [36]. These seemingly contradictory findings make the timing of a TERT promoter mutation unclear. It is also possible that there are different pathways by which glioblastoma can develop, leading to a common endpoint. It also points to our need to continue to understand the pathophysiology of glioblastoma development.

The role of TERT promoter mutation in the prognosis of patients suffering from glioma in general remains controversial and is nuanced. A series of studies have demonstrated that co-occurrence of the TERT promoter mutation with wild-type IDH is associated with a worse clinical outcome [37,38,39,40]. Interestingly, another group suggested that a TERT promoter was correlated with poor outcomes but only in patients with residual tumors who did not receive temozolomide [41]. Yet another group found that TERT promoter mutations did not correlate with improved or worsened outcomes [35]. These findings suggest that a TERT promoter mutation in those suffering from glioblastoma is likely associated with a worse prognosis, although in patients with a low-grade glioma, it is still unclear what role TERT promoter mutation has in prognostication. Given the prognostic implications of TERT mutation when associated with wild-type IDH, the mutation has also been included in the WHO 2021 integrated diagnosis for glioblastoma.

### 2.4. EGFR

Epidermal growth factor receptor (EGFR) is a family of integral membrane proteins belonging to the receptor tyrosine kinase family, involved in activating the key intracellular signal cascades RAS/MAPK, PI3K/AKT/mTOR, and JAK/STAT pathways, all of which are important in cellular proliferation [42]. There are four members of the EGFR family: ErbB1 (HER1), ErbB2 (HER2/Neu), ErbB3 (HER3), and ErbB4 (HER4) [43,44]. There are a number of ways in which EGFR can be dysregulated in tumor cells. The most common in glioblastoma is genomic amplification, which leads to overexpression of EGFR [45]. EGFR amplification is found in about 60% of patients with glioblastoma, resulting in tumorigenesis and tumor proliferation [46]. Given the number of extra receptors, normal regulation mechanisms are ineffective [43]. Another common mechanism of EGFR dysregulation is mutations to extracellular domains, leading to [47,48] unregulated activation of EGFR signaling and activation of downstream pathways.

Given the importance of EGFR in cellular proliferation and its critical role in many systemic malignancies, multiple inhibitors have been developed and used successfully in other diseases. First-generation EGFR inhibitors, such as erlotinib or lapatinib, which have shown efficacy in other malignancies, have not demonstrated improved survival in glioblastoma patients [43,49,50,51,52]. Later generations of EGFR inhibitors are still under investigation but have shown some promise in preclinical trials as well as some early-phase studies. One drug in particular, rindopepimut, showed significant promise in phase I and phase II trials but failed to demonstrate efficacy in phase III trials (results summarized in [53]). Currently, it is unclear why EGFR-targeted therapies are ineffective in glioblastoma. It could be due to potential genetic compensatory mechanisms, either by upregulation of the autocrine type of EGFR signaling or other compensatory pathways [43]. Regardless, the EGFR pathway continues to be an important potential therapeutic target. It may become more important as future therapeutic targets are developed to focus on which type of dysregulation applies for a particular patient’s glioblastoma, as there would be significantly different methods for targeting the different ways in which EGFR signaling can be dysregulated.

### 2.5. Gain of Chromosome 7/Loss of Chromosome 10

Another key genetic modification seen in glioblastoma and included within the 2021 WHO classification of glioblastoma is the gain of chromosome 7 and loss of chromosome 10 (+7/−10) [6], found in approximately 75% of glioblastoma and associated with a worse prognosis [54]. There are many hypotheses for this association. Firstly, loss of chromosome 10 is associated with a loss of tumor suppressor genes, which would predispose cells to tumorigenesis [55]. Additionally, aneuploidy is associated with worse survival in patients with astrocytoma [56], suggesting a +7/−10 mutation is indicative of significant genetic dysfunction. Further examination into the role of +7/−10 may be indicated to further elucidate potential etiologies of its importance in glioblastoma pathogenesis, though this is complicated given the numerous genetic and epigenetic changes associated with large-scale chromosomal abnormalities.

### 2.6. ATRX

ATRX is a member of the SWI/SNF superfamily of chromatin remodeling proteins that has important roles in maintaining heterochromatin and loading histones onto telomeres [57,58,59,60,61,62]. It also has known roles in cortical development [59] and is associated with the alternative lengthening of telomeres, through which cells lengthen telomeres without telomerase [63,64]. Although ATRX is common in low-grade glioma, it is much less commonly found in glioblastoma patients (about 10%) and has been associated with improvement in survival and increased radiosensitivity in patients with glioblastoma [65,66,67,68]. When interpreting prior studies evaluating the prognosis of ATRX mutations, one must be cautious given the association with IDH mutations. However, further study of the prognostic implications of ATRX is necessary, and investigations of ATRX and related pathways as a target are ongoing, given its implication in increasing sensitivity to DNA-damaging agents [63,65,69,70].

### 2.7. P53

TP53 is the most common mutated gene in all of human cancers [71], while the greater signaling pathway involving P53 is dysregulated in about 84% of glioblastomas [72]. P53 is at the center of the cell stress signaling pathway and is important in cell survival and apoptosis [71]. Mutations in P53 are associated with increased stem cell-like behavior of glioblastoma cells [73], increased migratory behavior/invasion [74], and progression of glioblastoma [75]. So far, no therapy targeting the P53 pathway has shown promise in humans, but there is some evidence of potential efficacy in animal models [72]. Given the integral role of P53 in many cellular processes, it seems unlikely that targeting P53 or the pathway directly is likely to yield significant therapeutic benefits without substantial off-target effects, though targeted delivery to tumor cells is an active field of development (reviewed extensively by Hsu et al. [76]).

### 2.8. RNA Sequencing

As previously mentioned, diagnosis of glioblastoma was classically performed through histological evaluation [5]. The WHO 2021 guidelines added critical genetic modifications to the diagnosis to create what is likely to be more prognostically accurate groups [6]. However, gross genomic mutations do not alone explain the diversity of clinical responses that patients experience, likely given the significant tumor heterogeneity [77]. Given this diversity, RNA sequencing of glioblastoma is being leveraged to better understand the tumor microenvironment, which comprises tumor, immunological, and stromal cells. A seminal paper in the field was published in *Cancer Cell* in 2010 and jumpstarted the rapidly progressing field of glioblastoma transcriptomics [78]. The researchers performed bulk RNA sequencing and, through clustering analyses, demonstrated four distinct molecular subtypes (proneural, neural, classical, and mesenchymal), each with disparate prognosis in the context of standard-of-care therapy. Unfortunately, this study was performed in the context of histopathologic diagnosis of glioblastoma, and these subtypes are based on the presence of *IDH* mutation and are therefore no longer used clinically, though novel classifications of the tumor microenvironment with bulk RNA sequencing continue to be evaluated [79].

Single-cell RNA sequencing is a next-generation sequencing technology that provides RNA sequencing at a single-cell level [80]. Glioblastoma is described as a heterogeneous tumor [77,81]; thus, widespread use of single-cell RNA sequencing can better characterize the tumor microenvironment and provide a better understanding of glioblastoma pathogenesis, treatment resistance, and novel therapeutic targets. The technology provides a massive amount of data, and research is ongoing on the best computational pipelines to analyze this data; however, one classification system that classifies glioblastoma tumor cells into four subtypes (NPC-like, MES-like, OPC-like, and AC-like) is widely utilized [82]. Single-cell RNA sequencing is now being used to better understand mechanisms of tumor progression, and one study evaluating the tumor microenvironment found that tumor recurrence is associated with decreased microglia, increased vascular endothelial growth factor expression, and increased blood–brain barrier permeability [83]. While the field is currently obtaining massive amounts of data from single-cell RNA sequencing to better understand the tumor microenvironment, as of yet, it is unclear what impact these data have on prognostication and therapeutic applications.

In theory, more advanced sequencing, such as single-cell RNA sequencing, may allow a better understanding of glioblastoma pathogenesis and treatment resistance as well as potential new or patient-specific therapeutics.

There are multiple other genetic mutations that can be associated with glioblastoma but have low frequency. Given the rapid advances in next-generation sequencing in both transcriptional activity and epigenetic regulation, our understanding of the dysfunctional pathways leading to glioblastoma formation will likely continue to improve, potentially leading to novel and effective therapies for patients afflicted with this terrible disease (see Table 1 for a brief synopsis of the discussed genetic mutations).

## 3. Noninvasive Diagnostic Testing for Glioblastoma

While the field of neuro-oncology has made significant advancements in prognosticating survival for patients with glioblastoma, initial diagnosis and disease monitoring remain areas with fewer breakthroughs [84]. The current gold standard for diagnosing glioblastoma involves obtaining a molecular profile of the tumor, which requires surgically acquired pathologic tissue for analysis. This approach allows for an understanding of the tumor’s genetic and molecular characteristics, critical for prognosis and personalized treatment. However, this reliance on invasive procedures poses challenges for both initial diagnosis and ongoing disease monitoring. After initial treatment, monitoring glioblastoma progression is primarily conducted through serial brain MRI scans. Unfortunately, such radiographic monitoring is suboptimal, as confounding diagnoses include radiation necrosis, pseudoprogression, and pseudoresponse [85]. Radiation necrosis typically occurs within 3 to 12 months following radiotherapy and manifests as increased post-contrast enhancement on conventional MRI [86]. Pseudoprogression, an exaggerated response to treatment, generally appears within 3 to 6 months post-radiotherapy and may occur with or without chemotherapy [87,88]. Pseudoresponse results from decreased vascularity due to treatment with the anti-angiogenic drug bevacizumab, leading to reduced vascular enhancement in an otherwise viable tumor [85,88]. These diagnostic challenges highlight the need for more reliable and noninvasive methods, which have become critical areas of research aiming to bridge the gap between the limitations of current imaging and the precision of molecular diagnostics.

As mentioned, the only method to definitively obtain an initial diagnosis or evaluate radiographic changes is through surgical specimens, which are inevitably associated with surgical risk. However, emerging noninvasive diagnostic techniques are gaining attention as promising alternatives for glioblastoma diagnosis. This section of the literature review will explore these up-and-coming methods, focusing on their strengths, limitations, and potential to shape the future of glioblastoma diagnostics.

### 3.1. Liquid Biopsy

Traditionally, diagnosis of glioblastoma is performed on tissue obtained directly from the primary tumor through resection or stereotactic biopsy. In contrast, liquid biopsy involves the analysis of bodily fluids, such as blood obtained via peripheral venipuncture or cerebrospinal fluid (CSF) collected through lumbar or cisternal puncture, offering a relatively noninvasive alternative to direct tissue sampling [89]. Liquid biopsy techniques enable the detection and monitoring of glioblastoma by analyzing circulating tumor cells (CTCs), extracellular vesicles (EVs), and circulating tumor DNA (ctDNA) [90].

CTCs are intact cells that have detached from the primary tumor and have entered the bloodstream, exposing them to potential sampling [91]. These cells are difficult to compare given a lack of a standard method for isolation and characterization as well as the rarity of metastatic spread in glioblastoma [90]. MacArthur et al. observed a decrease in CTC detection rate in patients post-radiotherapy; however, the small sample size is a limitation of the study [92].

EVs are small, lipid bilayer-bound particles that exist predominantly in the extracellular space, encapsulating components from their cell of origin, including DNA, RNA, and proteins [93]. An advantage of testing for EVs is that the biomolecules within them are surrounded in a lipid bilayer, which protects them from enzyme degradation in the extracellular medium, allowing them to cross an intact blood–brain barrier (BBB) [94]. Some studies have shown increased concentration of EVs in glioblastoma patients when compared to healthy individuals [95], with a decrease being related to treatment and an increase being related to tumor recurrence [96].

ctDNA consists mainly of short DNA fragments, approximately 140–180 base pairs in length, that have separated from the tumor and circulate within the blood or CSF [97]. Although theoretically promising, ctDNA has poor sensitivity in patients with glioblastoma. In an analysis of 640 patients with a range of malignancies, Bettegowda et al. found ctDNA, readily detectable in most systemic malignancies, was observable in less than 10% of patients [98,99]. Importantly, there is a suggestion that liquid biopsy sensitivity may be improved in testing CSF, though this sensitivity may be anatomically constrained to tumors near a CSF interface [100]. To circumvent this constraint, Furey et al. implanted reservoirs into the resection cavity at the time of initial resection for serial, noninvasive monitoring, with promising preliminary results [101]. Extensions of such trials are necessary to prove efficacy [102].

The challenge of detecting ctDNA in plasma versus CSF is hypothesized to be due, in part, to the BBB. Whether the BBB restricts the passage of ctDNA into the bloodstream is still controversial [99]. One study suggested that the BBB prevents glioma-derived ctDNA from crossing into the blood plasma [103], while another study demonstrated that BBB disruption caused by brain tumors increases the ctDNA plasma concentration and detection specificity [104]. Recent advancements have demonstrated that the use of focused ultrasound (FUS) can improve ctDNA detection. In glioblastoma models involving mice and pigs, FUS has been shown to increase ctDNA levels, enabling improved sensitivity of biomarkers such as EGFR and TERT [105] as well as enhanced detection of green fluorescent protein (eGFP) [106]. A human trial further corroborated these findings, showing similar success without inducing detectable tissue damage or significant inflammation or immune response within a short time frame [107].

When it comes to liquid biopsy, CSF, despite being more invasive, has historically demonstrated greater sensitivity than blood plasma for diagnosis of brain tumors. Further study is necessary to improve the sensitivity and efficacy of both methods of sampling, and larger samples are required. Further work will have to be conducted to understand the potential benefits and, importantly, the potential limitations of liquid biopsy.

### 3.2. MR Spectroscopy

Magnetic resonance spectroscopy (MRS) is a noninvasive imaging modality that complements conventional magnetic resonance imaging (MRI) by producing a volume pixel (voxel) to represent the concentration of various metabolites in the tissue [108]. Concentration of key metabolites, such as creatine (Cr), phosphocholine (Pcho), N-acetylaspartate (NAA), and lactate, can be ascertained via MRS [109]. Total Cr can be used to characterize tumor type and grade, while lactate is often increased in a high-grade glioma [110]. Clinically, MRS has applications in tumor diagnosis and grading, distinguishing between tumor types, assessing therapeutic response, and detecting recurrence [111].

Historically, contrast-enhancing lesions observed on MRI have been attributed to tumor progression or recurrence. However, patients who have undergone radiotherapy and chemotherapy treatments may also exhibit similar findings on MRI [112]. Such observations may result from post-treatment radiation effects, recurrent tumor tissue, or a combination of both, which conventional MRI is unable to distinguish [113]. MRS, however, has demonstrated utility in differentiating tumor types, grading, treatment response, and recurrence [114,115,116]. Elias et al. examined tumor recurrence and post-radiation effects and found that an elevated non-normalized choline-to-N-acetylaspartate (Cho/NAA) ratio was significantly associated with tumor recurrence, even after adjustments for multiple comparisons with 86% sensitivity and 90% specificity [117]. Others have highlighted the value of metabolic concentration ratios in distinguishing glioblastoma from solitary brain metastasis [118].

Magnetic resonance spectroscopy can help distinguish glioblastoma from lower-grade gliomas by detecting metabolic markers such as 2-hydroxyglutarate (2HG) [119]. The presence of 2HG, associated with IDH mutations [120], is indicative of lower-grade gliomas, while glioblastoma is an IDH wild-type [6,121,122]. This distinction is clinically significant for prognosis, as IDH mutations are linked to improved survival outcomes [123]. Furthermore, 2HG peaks are observed exclusively in tumorous tissue and not in normal tissue [122,124]. However, tumor volume can affect the sensitivity of 2HG detection, as volumes below the lower MRS resolution limit of approximately 8 mL may hinder accurate measurement [125]. A study found 100% specificity and sensitivity in their detection of 2HG, while another had a sensitivity as low as 8% for small volume tumors (<3.4 mL) [124,125].

Thus, the primary application of MRS is its capacity to differentiate between tumor progression and post-therapy changes resulting from chemotherapy or radiation [126]. It can also be used to help distinguish between gliomas with and without IDH mutations, which has significant prognostic value. However, MRS has certain limitations, including a relatively long acquisition time (typically over 10–15 min) and user-dependent placement of voxel sampling [110,127]. An additional challenge is “voxel bleeding”, which can lead to signal contamination from adjacent tissues. This issue can be mitigated by only selecting voxels where the tissue of interest comprises at least two-thirds of the voxel [128].

### 3.3. Relative Cerebral Blood Volume and Fractional Tumor Burden

Relative cerebral blood volume (rCBV) is an advanced MRI sequence derived from dynamic susceptibility contrast (DSC) MRI, a perfusion-weighted imaging technique that utilizes a contrast to quantify blood volume [129,130]. This metric can then characterize tumor grade, aggressiveness, and monitor treatment response or recurrence [131,132]. Glioblastoma, known for its high vascularity due to extensive angiogenesis, typically exhibits elevated rCBV values. rCBV analysis generally focuses on three key regions: the solid tumor area, the peritumoral area, and the region adjacent to the peritumoral area [133]. These areas are important in the evaluation of the tumor type as well as distinguishing tumor recurrence or post-treatment radiation necrosis.

Previously, it has been shown that rCBV measurements in the peritumoral edema of lesions can differentiate glioblastoma from metastatic tumors [132,133,134]. Tumor recurrence often induces angiogenesis, leading to the development of abnormal blood vessels with increased permeability, which results in elevated rCBV values relative to normal brain tissue [135]. Additional studies have found that rCBV, along with mean, maximum, and minimum peak height measurements, are useful for distinguishing recurrent glioblastoma from radionecrosis [130,136]. Schmainda et al. demonstrated that a decrease in rCBV, measured at either 2 or 16 weeks following treatment initiation with bevacizumab, was predictive of a clear improvement in overall survival of patients [137].

A common limitation in rCBV measurements arises in multicenter trials where standardized imaging protocols are lacking [138]. User-dependent error can occur when measuring rCBV by selecting the “hot spot” of the tumor, as it does not account for the entire volume or the heterogeneity of the tumor [129,139]. For instance, Smits et al. published a multicenter study that reported repeatability of about 50% and reproducibility of 5.5% in rCBV results, underscoring the variability that can result from inconsistent imaging procedures [140]. This limitation can be minimized by adhering to standardized imaging parameters, including consistent contrast administration protocols [141]. Standardization of rCBV hs been shown to improve inter- and intra-patient consistency and repeatability compared with manually defined normal regions of interest [142]. Another important limitation in rCBV assessment is the issue of “leakage” in cases where the blood–brain barrier is compromised, which can make it more difficult to interpret imaging and should be accounted for and corrected during analysis [143]. Overall, rCBV is a useful imaging modality that can be helpful in adding further evidence when deciding next steps in patient care, in particular when looking for recurrence or treatment effect where other imaging modalities often fall short. The lack of standardization in protocols has been a weakness, but with further standardization, the utility of rCBV will likely continue to increase.

Building upon the utility of rCBV, fractional tumor burden (FTB) mapping further enhances diagnostic capabilities by providing a voxel-wise assessment of prefusion across the entire lesion. FTB maps use normalized rCBV values and predefined thresholds to spatially differentiate the extent of abnormal tissue within a region from treatment-related effects, making it more informative for assessing tumor heterogeneity and guiding treatment decisions [144]. A key advantage of FTB maps is their ability to provide perfusion characteristics of the entire lesion through per-voxel measurements of cerebral blood volume rather than relying on a single averaged value to represent the lesion’s perfusion [139]. This approach reduces operator dependence associated with the traditional “hot spot” ROI method, thereby enhancing the reproducibility and reliability of the measurements [139,145]. When combined with MRS, FTB maps hold the potential to become sufficiently reliable, possibly eliminating the need for invasive biopsy sampling in the future [146].

A limitation of studying the effectiveness of FTB maps is the correlation of histopathologic results for every voxel of the contrast-enhancing volume of the map [139]. Because of the heterogenic nature of glioblastoma, diagnosis of samples from separate locations may differ, as even diagnostic agreement among pathologists can vary with a given tissue sample [147]. Overall, FTB mapping has the potential to improve diagnostic accuracy and has been a useful tool in determining the optimal place to obtain a stereotactic biopsy. As we continue to use these tools and others, it may become even more useful to separate treatment effect from tumor for other potential invasive or noninvasive interventions.

### 3.4. Combination FET PET/MRI

As has been discussed before, the diagnostic accuracy of conventional follow-up MRI in glioblastoma is limited due to treatment-related signal changes [148], which make it difficult to distinguish between tumor progression and radionecrosis [149]. To improve diagnostic accuracy, molecular imaging with amino acid PET tracers in combination with MRI has been explored [150]. One specific amino acid analog used for PET/MRI is fluoro-ethyl-tyrosine (FET), a synthetic amino acid derived from tyrosine [151]. Like natural tyrosine, FET is passively taken into cells in exchange for leucine [152]. This transport mechanism is often overexpressed in brain tumor cells due to increased amino acid demand for protein synthesis, rapid cellular proliferation, and angiogenesis [153,154]. After entering the cell, FET is not further metabolized, resulting in higher retention time in pathologic tumor tissue than natural amino acid radiotracers [152].

FET offers a few advantages when compared to other radiotracers [155]. 18F-FDG, unlike FET, is a glucose analog, so it is taken up by all cells with high glucose metabolism. This is a drawback in brain imaging because normal brain tissue has naturally high glucose metabolism, leading to a high background uptake, making it hard to distinguish tumors [156].

A study by Galldiks et al. demonstrated that the combined use of static and dynamic FET PET with MRI achieved an accuracy of 96% in distinguishing pseudoprogression from true tumor progression [157]. Other studies have shown high diagnostic accuracy with FET PET in differentiating radiation-induced injury from tumor progression in patients with brain metastases [158,159,160]. Notably, FET PET/MRI appears to offer an advantage over rCBV, providing clearer tumor-to-background contrast than rCBV maps [161]. This finding appears to align with studies showing that FET PET reveals a larger tumor volume of metabolic activity compared to contrast-enhanced MRI, which had been validated with stereotactic biopsy [162,163].

The integration of PET with MRI offers an advantage in differentiating between tumor recurrence and treatment-related changes, such as radiation necrosis or pseudoprogression, which often present with overlapping radiographic features on MRI alone [84]. The addition of a radiotracer in PET/MRI imaging provides a less invasive and more cost-effective alternative to stereotactic biopsy for accurate diagnosis or treatment management. However, logistical challenges, such as limited availability of radiotracers due to their short half-lives, may affect the feasibility of this technique in some clinical settings [84].

### 3.5. Amide Proton Transfer

Amide proton transfer (APT) imaging is an advanced molecular MRI technique derived from chemical exchange saturation transfer. APT enables the detection of endogenous mobile proteins and peptides, serving as an indicator of intercellular metabolic changes [164], and demonstrates promise in the diagnosis of glioblastoma. The technique utilizes targeted pulses to saturate amide protons in mobile proteins and peptides. As these saturated protons exchange with free water protons, the water signal in tissues is partially reduced. This signal alteration correlates with the concentration of mobile proteins and peptides, offering insights into cellular metabolic and pathophysiological changes [165]. In tumor-affected tissue, where protein and peptide levels are often elevated, there is an increase in the exchangeable amide protons, making APT imaging particularly valuable for detecting these biochemical changes [166].

There is evidence to suggest that APT imaging can effectively differentiate between low-grade and high-grade gliomas [166,167,168,169], likely attributable to increased cellular density [170]. In a comparison study with diffusion-weighted imaging, which is more classically used to assess cellularity, Bai et al. demonstrated that APT MRI offers superior imaging capabilities in assessing tumor cellularity [170].

Additionally, elevated APT values have been associated with more rapid disease progression and decreased survival in glioma patients [171]. In animal models, APT imaging has shown promise in differentiating radiation necrosis from active tumor tissue, though it requires further evaluation in patients [172]. Moreover, studies have indicated that unmethylated MGMT glioblastoma [173] tends to exhibit higher ATP values than WHO IV glioma or glioblastoma with methylated MGMT [174]. However, these associations have not been consistently demonstrated [175].

Importantly, APT has less operator variation than other novel advanced imaging [176]. However, as APT imaging detects endogenous mobile proteins, cystic regions within tumors can produce elevated APT signal intensities [177]. Overall, APT imaging is another modality that, when utilized, could improve diagnostic yield and help glean insight into the genetics of the patient’s glioma, though it is in its early phases of development and requires further study.

### 3.6. Radiomics

Radiomics is an emerging field that utilizes advanced computational techniques to extract and analyze quantitative features from medical images, enabling the identification of patterns through machine learning that are not possible to identify through other methods [178,179]. The advancement of radiomics is inherently multidisciplinary, involving collaboration among clinicians, molecular biologists, statisticians, and bioengineers [179].

Radiomics aims to facilitate a wide range of clinical applications by assimilating all radiologic data to predict prognosis and noninvasively predict genetic markers, such as IDH mutation and MGMT promoter methylation status. Both IDH mutation and MGMT promoter methylation serve as prognostic and predictive biomarkers, associated with improved survival (see above) [23,89]. Radiomic models can also incorporate higher-order radiomic features to predict survival by combining features such as contrast enhancement, the distance of a glioblastoma from the subventricular zone, and the degree of associated mass effect [178]. Notably, a fully automated radiomics model has demonstrated the ability to distinguish between high-grade and low-grade gliomas with 90% accuracy [180].

Radiomics may be leveraged to predict response to clinical outcomes. In a study examining the role of radiomics in identifying patients likely to benefit from anti-angiogenic therapies, an area under the receiver operating characteristic curve of 0.792 was achieved for predicting overall survival in patients treated with bevacizumab [181]. Additionally, radiomics-based approaches have shown approximately 80% sensitivity in distinguishing tumor progression from pseudoprogression [182,183]. Another study highlighted the utility of tumor shape features and surface irregularities in differentiating glioblastoma from pseudoprogression [184]. These findings present the potential of radiomics in glioblastoma diagnosis and treatment planning, including its role in assisting with radiotherapy target volume segmentation and measuring and predicting treatment response [185].

Radiomics has the potential to transform glioblastoma management by enabling a personalized approach at various stages of disease progression as well as tracking improved treatment efficacy [186,187]. However, its development and implementation face several limitations. One challenge in constructing radiomic models is the difficulty of validating findings against the “gold standard” tissue biopsy due to the requisite imprecision, though minimal, of image-guided biopsy [188]. Additionally, multicenter studies often utilize varied imaging protocols, including differences in slice thickness, resolution, and washout periods, which complicate model standardization [189]. MRI-specific factors, such as field strength, pulse sequences, and manufacturer differences, have also been shown to impact radiomic outcomes and reinforce the need for standardized protocols to ensure model accuracy and reliability [190,191].

Overall, as imaging technology and other less-invasive diagnostic tools continue to progress, we will be able to better diagnose and surveil treatment and disease recurrence (see Table 2 for a brief synopsis of the discussed less-invasive diagnostic tools).

## 4. Conclusions

Glioblastoma remains one of the most vexing malignancies, given we have not pushed survival curves significantly in decades despite continued improvement in understanding of the disease process. In this review, we reviewed some of the most commonly mutated genes associated with glioblastoma as well as potential less-invasive techniques one may use to diagnose glioblastoma or assess the efficacy of therapy. As we continue to advance in our genetic and imaging techniques, one can be hopeful that we will continue to improve our understanding of glioblastoma and potentially find better therapeutic interventions as well as noninvasive techniques to diagnose patients with glioblastoma and then follow along as their treatment progresses.

## Figures and Tables

**Table 1 cancers-17-00124-t001:** Synopsis of the major genetic mutations found in glioblastoma, the most common etiology of the mutation, the diagnostic technique to uncover the mutation, as well as the clinical significance.

Gene	Mutation	Diagnostic Technique	Clinical Significance
Isocitrate dehydrogenase	R132H is the most common; other mutations are present but are less common	ICH for R132H; sequencing for other mutations	IDH mutations are associated with improvements in survival
O6-methylguanine-DNA methyltransferase (MGMT)	Epigenetic silencing	Promoter methylation, although MGMT activity can also be assessed	Decreased MGMT activity is associated with better response to temozolomide
Telomerase reverse transcriptase (TERT)	Promoter mutation	Sequencing	TERT promoter mutation in association with wild-type IDH is associated with worse clinical outcomes
Epidermal growth factor receptor (EGFR)	Amplification, extracellular domain mutations	Sequencing	Mixed, could be potential therapeutic targets, although not currently effective
Chromosome 7 and 10	Gain of chromosome 7 and loss of chromosome 10	Karyotype, sequencing	Worse prognosis
Alpha-thalassemia/mental retardation (ATRX)	Genomic mutation at active domain	Sequencing	Potential improvement in survival due to increased radiosensitivity
P53	Genomic mutation	Sequencing	Increased aggressive behavior of glioblastoma cells

**Table 2 cancers-17-00124-t002:** Synopsis of the major advanced diagnostic techniques used to less-invasively or noninvasively diagnose those with glioblastoma or surveil treatment response/recurrence.

Modality	Diagnostic Use	Advantages	Limitations
Liquid biopsy	Detects tumor-specific biomarkers (ctDNA, CTC, EVs, etc.) in blood or CSF	Blood or CSF samples can provide molecular profiling and track treatment response	May not localize the tumor and needs further work to understand possible clinical applications.
Magnetic resonance spectroscopy	Analyzes chemical composition (Cr, Pcho, NAA, etc.) within the tumor using MRI	Detects metabolic changes and differentiates tumor progression from treatment effect	Long acquisition time and user-dependent placement of voxel sampling
Relative cerebral blood volume and fractional tumor burden	Measures vascular density and tumor burden proportion using perfusion-weighted MRI	Differentiates high-grade gliomas, identifies active tumor areas, and supports treatment monitoring	Lack of imaging protocol can lower reproducibility between treatment centers
Combination FET PET/MRI	Combines PET imaging with FET tracer and MRI to detect amino acid metabolism	Differentiates progressive disease and treatment response with clearer tumor-to-background contrast than other modalities	Involves exposure to radioactive tracers
Amide proton transfer	Detects mobile proteins and peptides in tumors via saturation of amide protons in the peptide bonds using MRI	Provides information on metabolic activity and differentiation of glioma grade	Variability in signal intensity depending on tumor heterogeneity
Radiomics	Extracts complex data from imaging modalities to identify patterns and predict outcomes	Potential for personalized medicine and enables prediction of tumor behavior	Computationally intensive, requiring large datasets and standardization

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
