# Peer review of "Advances in Glioblastoma Diagnosis: Integrating Genetics, Noninvasive Sampling, and Advanced Imaging"

_cancers, 2025, doi:10.3390/cancers17010124_

Round 1
Reviewer 1 Report
Comments and Suggestions for Authors
In the manuscript, entitled „Advances in Glioblastoma Diagnosis: Integrating Genetics, Noninvasive Sampling, and Advanced Imaging” authors provide a review of the genetic mutations important in the pathogenesis, diagnosis, prognosis, and treatment of glioblastoma. Then they describe novel diagnostic tools that allow for faster and less invasive diagnosis, as well as monitoring of treatment response and disease recurrence. They hope, that this summary will serve as a reference for the current state-of-the-art and future directions for glioblastoma treatment.
In my opinion this review article contains the most important diagnostic tools for glioblastoma, especially some newly developed methods that are not in routin use by neuro-oncologists yet. Turning focus on these new methods forms the main benefits of the manuscript and may promote introducing supplementer diagnostic method worldwide.
I suggest to accept the manuscript in recent form.
Author Response
Comment 1:
In the manuscript, entitled „Advances in Glioblastoma Diagnosis: Integrating Genetics, Noninvasive Sampling, and Advanced Imaging” authors provide a review of the genetic mutations important in the pathogenesis, diagnosis, prognosis, and treatment of glioblastoma. Then they describe novel diagnostic tools that allow for faster and less invasive diagnosis, as well as monitoring of treatment response and disease recurrence. They hope, that this summary will serve as a reference for the current state-of-the-art and future directions for glioblastoma treatment.
In my opinion this review article contains the most important diagnostic tools for glioblastoma, especially some newly developed methods that are not in routine use by neuro-oncologists yet. Turning focus on these new methods forms the main benefits of the manuscript and may promote introducing supplementer diagnostic method worldwide.
I suggest to accept the manuscript in recent form.
Response 1:
Thank you for the comments – no revisions requested from this reviewer.
Reviewer 2 Report
Comments and Suggestions for Authors
The authors in this review have addressed some of the mutations frequently present in glioblastoma multiforme, as well as discuss a range of genetic and imaging methods that can be widely used by researchers in different fields in order to provide an early diagnosis along with the possibility of evaluating the efficacy of therapy. The authors provide a very thorough and detailed review in all its parts and address the topics with seriousness and comparison, as well as presenting some rather up-to-date citations. Overall, this review could improve the understanding of glioblastoma and provide ideas/updates on non-invasive techniques that could potentially be used in the diagnostic/medical, and also prognostic, field.
Minor revision:
- The review is well structured and easy for the reader, however, to streamline the amount of information presented, it is recommended to add one or two figures to make the work more appreciable.
- line 80 replace "ICH" with "IHC".
Author Response
Comment 1: The review is well structured and easy for the reader, however, to streamline the amount of information presented, it is recommended to add one or two figures to make the work more appreciable.
Response 1:
Thank you for the comments – we have attempted to consolidate the described details into two tables (common genes and novel diagnostic technologies). Illustrating additional novel figures is out of scope for the purposes of this review.
Comment 2: line 80 replace "ICH" with "IHC".
Response 2: We have made this correction.
Reviewer 3 Report
Comments and Suggestions for Authors
The manuscript Gough et al. characterizes advances in GBM diagnosis based on molecular (genomic and transcriptomic) and imaging data analysis developed in recent years. The authors discuss mutation criteria incorporated into the WHO glioma grading system in 2021, schemes of treatment, and prognostic outcomes. Given the constantly changing landscape of glioblastoma, the authors discuss the limitations of current diagnostic methods and emphasize the need for developing less invasive procedures to monitor tumor progression and intratumoral alterations on molecular and metabolic levels. Overall, the manuscript is well-designed and provides scientifically relevant information. To improve manuscript quality and interest to a broader audience I have a few recommendations:
1. Recent research has highlighted the critical role of the tumor microenvironment, particularly tumor-infiltrated myeloid-derived cells, in glioblastoma progression. Increasing evidence shows that the high ratio of neutrophils to lymphocytes (NLR) in CBC tests is prognostic for cancer severity. This topic is underrepresented and needs to be expanded by the authors (probably as a small chapter like the Role of microenvironmental biomarkers in glioblastoma diagnosis and progression).
2. It would be helpful to include an overview of clinical trials relevant to the path of treatment depending on diagnostic criteria. Relatedly, the authors mentioned the IDH1 R132 mutations and the corresponding path of treatment and overall prognosis. Are there any other molecular abnormalities that relate to the choice of participation in clinical trials? It will be nice to have a small chapter related to this topic.
3. The correlation between environmental factors and neuroblastoma subtypes at the moment of initial diagnostics would be also of great interest to readers.
I recommend the manuscript for publication in Cancers MDPI journal with minor revision.
Author Response
Comment 1: Recent research has highlighted the critical role of the tumor microenvironment, particularly tumor-infiltrated myeloid-derived cells, in glioblastoma progression. Increasing evidence shows that the high ratio of neutrophils to lymphocytes (NLR) in CBC tests is prognostic for cancer severity. This topic is underrepresented and needs to be expanded by the authors (probably as a small chapter like the Role of microenvironmental biomarkers in glioblastoma diagnosis and progression).
Response 1:
We agree – the tumor microenvironment is a rapidly evolving field. Today, the best understanding of the glioblastoma tumor microenvironment has been through single cell RNA sequencing data given the significant intra-tumoral heterogeneity. We have expanded on this section and cited seminal papers on the subject. The clinical application of NLR in glioblastoma is not currently evaluated, particularly given the known systemic CD4 depletion; given the complexity of this subject, further discussion is out of scope of this review.
Comment 2: It would be helpful to include an overview of clinical trials relevant to the path of treatment depending on diagnostic criteria. Relatedly, the authors mentioned the IDH1 R132 mutations and the corresponding path of treatment and overall prognosis. Are there any other molecular abnormalities that relate to the choice of participation in clinical trials? It will be nice to have a small chapter related to this topic.
Response 2:
Currently, clinical trials for glioblastoma are limited. With the recent change in WHO classification, the IDH mutation is no longer classified as glioblastoma – therefore, the targeted therapy vorasidenib was limited to grade 2 glioma. MGMT status is a significant factor in eligibility for clinical trials, as unmethylated MGMT implies a patient is a poor candidate for standard of care therapy (temozolomide) and therefore could benefit from novel therapies (this has been made more clear in the manuscript). There are not other consistent abnormalities that make patients eligible for trials, though individual trials may target specific mutations (e.g. EGFRvIII as we described in the manuscript)
Comment 3: The correlation between environmental factors and neuroblastoma subtypes at the moment of initial diagnostics would be also of great interest to readers.
Response 3:
The authors believe the reviewer meant glioblastoma subtypes, however, environmental correlations of glioblastoma subtypes are not accepted at this time. However, we have expanded the discussion of glioblastoma subtypes within the RNA sequencing section – molecular subtypes are thought to have potential prognostic implications.